# Synaptopodin-2 Isoforms Have Specific Binding Partners and Display Distinct, Muscle Cell Type-Specific Expression Patterns

**DOI:** 10.3390/cells13010085

**Published:** 2023-12-30

**Authors:** Keerthika Lohanadan, Marvin Assent, Anja Linnemann, Julia Schuld, Lukas C. Heukamp, Karsten Krause, Matthias Vorgerd, Jens Reimann, Anne Schänzer, Gregor Kirfel, Dieter O. Fürst, Peter F. M. Van der Ven

**Affiliations:** 1Institute for Cell Biology, University of Bonn, 53121 Bonn, Germany; 2Department of Pathology, University Hospital Bonn, 53127 Bonn, Germany; 3Department of Neurology, Heimer Institute for Muscle Research, University Hospital Bergmannsheil, Ruhr-University Bochum, 44789 Bochum, Germany; 4Department of Neurology, Neuromuscular Diseases Section, University Hospital Bonn, 53127 Bonn, Germany; 5Institute of Neuropathology, Justus-Liebig-University Giessen, 35392 Giessen, Germany

**Keywords:** synaptopodin-2/SYNPO2/myopodin, striated muscle, smooth muscle, alternative splicing, protein isoforms, synemin, α-actinin, nemaline myopathy, neurogenic atrophy, denervation

## Abstract

Synaptopodin-2 (SYNPO2) is a protein associated with the Z-disc in striated muscle cells. It interacts with α-actinin and filamin C, playing a role in Z-disc maintenance under stress by chaperone-assisted selective autophagy (CASA). In smooth muscle cells, SYNPO2 is a component of dense bodies. Furthermore, it has been proposed to play a role in tumor cell proliferation and metastasis in many different kinds of cancers. Alternative transcription start sites and alternative splicing predict the expression of six putative SYNPO2 isoforms differing by extended amino- and/or carboxy-termini. Our analyses at mRNA and protein levels revealed differential expression of SYNPO2 isoforms in cardiac, skeletal and smooth muscle cells. We identified synemin, an intermediate filament protein, as a novel binding partner of the PDZ-domain in the amino-terminal extension of the isoforms mainly expressed in cardiac and smooth muscle cells, and demonstrated colocalization of SYNPO2 and synemin in both cell types. A carboxy-terminal extension, mainly expressed in smooth muscle cells, is sufficient for association with dense bodies and interacts with α-actinin. SYNPO2 therefore represents an additional and novel link between intermediate filaments and the Z-discs in cardiomyocytes and dense bodies in smooth muscle cells, respectively. In pathological skeletal muscle samples, we identified SYNPO2 in the central and intermediate zones of target fibers of patients with neurogenic muscular atrophy, and in nemaline bodies. Our findings help to understand distinct functions of individual SYNPO2 isoforms in different muscle tissues, but also in tumor pathology.

## 1. Introduction

Synaptopodin-2 (SYNPO2, myopodin) is a member of the podin protein family, together with synaptopodin (SYNPO) and synaptopodin 2-like (SYNPO2L). SYNPO2 was initially described as fesselin, an actin-binding, synaptopodin-like protein purified from chicken gizzard that is highly expressed in smooth and striated muscle tissues. Other tissues, such as kidney and brain, express only minor amounts of the protein [1]. Later studies revealed that fesselin is, in fact, avian SYNPO2 [2]. SYNPO2 is also known as genethonin-2 [3], the product of a gene that was found to be downregulated in patients with Duchenne muscular dystrophy (DMD) or as myopodin [4]. In the latter work, it was confirmed that the protein is expressed in all types of muscle cells, and localizes to Z-discs in striated muscle cells, while in smooth muscle, SYNPO2 localizes to dense bodies [5]. In C2C12 myoblasts, SYNPO2 was claimed to redistribute between nuclei and the cytoplasm, depending on the state of differentiation of the myocytes and on stress [4]. The nuclear–cytoplasmic shuttling was found to be regulated by phosphorylation: only phosphorylated SYNPO2 binds 14-3-3 proteins, which in its turn enables the binding of importin α and thus nuclear import of SYNPO2 [6,7].

The *SYNPO2* gene was initially described as an intronless gene in the mouse [4]. However, later studies revealed that differential splicing and alternative usage of transcriptional start sites in fact enable the expression of several different SYNPO2 isoforms (Figure 1) [8,9,10,11,12]. Interestingly, three of the proposed isoforms contain an amino-terminal extension with a PDZ domain, the only structured domain in the otherwise natively unfolded protein. This PDZ domain directly interacts with VPS18 [13], linking SYNPO2 to a protein complex involved in autophagosome formation. Alternative splicing would allow the generation of SYNPO2 variants with three different carboxy-termini [8,9,11] that influence the effect of the individual proteins on actin filament organization [11]. By contrast, the PDZ domain is not directly involved in the localization of SYNPO2 along actin filaments [8].

The major part of SYNPO2, encoded by the largest part of exon 4, is present in all thus far known SYNPO2 isoforms (Figure 1). This part contains most of the binding sites that seem to be responsible for the localization of SYNPO2 to myofibrillar Z-discs in striated muscles. SYNPO2 not only binds and bundles actin filaments [1,4,14], but also interacts with α-actinin and filamin C, two well-known Z-disc components [7,14]. The early expression of SYNPO2 in differentiating cultured skeletal myotubes, even before appearance of α-actinin-2, the striated muscle-specific isoform of α-actinin, points to an important structural function of the protein in the reorganization of the actin cytoskeleton during the assembly of the contractile apparatus [14].

The preeminent importance of SYNPO2 is underlined by a number of genetic findings: *Synpo2* was reported to be a novel lethal gene in mice by the International Mouse Phenotyping Consortium (IMPC) [15]. This group developed approximately 5000 knockout models in mice, of which 410, including that of *Synpo2*, were found to be lethal. The phenotype of mice homozygous for the Synpo2^tm1b(EUCOMM)Wtsi^ allele was described as preweaning lethality with complete penetrance, indicating that SYNPO2 is crucial for mouse embryonic development. In humans, *SYNPO2* variants were presented as candidates of monogenic nephrotic syndrome and proteinuria [16,17]. In addition, SYNPO2 was described to play a role in several aspects of tumor development (for review, see [18]). The nuclear function of SYNPO2 was suggested to be involved in tumor suppressor activity, whereas cytoplasmic SYNPO2 might have a tumor activator role [19]. It is evident that a precise knowledge of the expression patterns of SYNPO2 isoforms, and the function of the different amino- and carboxy-terminal parts of these isoforms, are pivotal for a better understanding of the role of SYNPO2, not only in striated and smooth muscle development and function, but also in many forms of cancer. We present here the differential isoform expression in striated muscles and tissues containing a large proportion of smooth muscle cells. Furthermore, we identify novel interaction partners for the parts of SYNPO2 that are specific for a subset of isoforms.

## 2. Materials and Methods

### 2.1. Patients and Human Tissue Specimens

In this study, skeletal muscle samples were collected from adult patients without muscle pathology, as well as from patients with neurogenic muscular atrophy or nemaline rod myopathy. The patients with neurogenic muscular atrophy were part of a previous proteomic study that analyzed the composition of the central and intermediate zones in target fibers (patient IDs 1 and 16 in Table 1 in ref. [20]). From the controls and these male patients (aged 51 and 61 years at biopsy), gastrocnemius muscle biopsies were analyzed. From two patients with nemaline rod myopathy, muscle biopsies were obtained from quadriceps femoris muscle (male aged 54 years, and female aged 67 years at biopsy). In both biopsies, the presence of nemaline bodies was confirmed using a modified Gomori trichrome technique [21]. Cardiac tissue was obtained from the left ventricle of an explanted heart from a 15-year-old child with acute rejection reaction after heart transplantation at the age of 2 years. Normal uterus and prostate tissue was obtained from the BioMaSoTa-Biobank of the Department for Integrated Oncology (CIO) Bonn, Germany. Colon tissue was from a biopsy of a patient with Hirschsprung disease. All material was collected independent of and prior to our study for diagnostic purposes. Use of the material was approved by the ethical committees of the Universities of Cologne/Bonn (#13-091), the Ruhr-University Bochum (#4368-12) and the University of Giessen (#AZ07/09). Written informed consent was obtained from all subjects involved in this study.

### 2.2. RT-PCR

For a semi-quantitative analysis of SYNPO2 isoform expression in various human tissues at the RNA level, total RNA was isolated from tissue specimens using the RNeasy Fibrous Tissue Mini Kit following the manufacturer’s instructions (Qiagen, Hilden, Germany). cDNA was prepared using random nonamers and the FIREScript RT cDNA Synthesis Kit (Solis Biodyne, Tartu, Estonia) or the Omniscript RT Kit (Qiagen, Hilden, Germany). cDNA was amplified by PCR using the oligonucleotides listed in Table 1 using FIREPol Master Mix Ready to Load (Solis Biodyne). Oligonucleotides were designed using OligoPerfect Primer Designer (https://www.thermofisher.com/de/de/home/life-science/oligonucleotides-primers-probes-genes/custom-dna-oligos/oligo-design-tools/oligoperfect.html (accessed on 10 January 2023)) by submitting the sequence of the regions flanking the individual exon boundaries. Fragments were analyzed by agarose gel electrophoresis. Gels were photographed and analyzed by densitometry using a GelDoc XR Imaging system and Quantity One 4.6 software (Bio-Rad, Feldkirchen, Germany).

### 2.3. Preparation of Tissue Extracts and Western Blotting

Comparative protein expression analyses were performed essentially as described [22]. In brief, frozen tissue samples were weighed, mechanically disrupted using a TissueLyser LT (Qiagen, Hilden, Germany) at 50 Hz, and dissolved in 15 μL urea buffer [2 M thiourea, 7 M urea, 5 mM EDTA, 1 mM DTT and protease inhibitors (P8340; Sigma-Aldrich Chemie, Taufkirchen, Germany.) in 100 mM Tris pH 8.6] per 1 mg of tissue sample by homogenization for 3 min at 50 Hz. Preheated SDS sample buffer was added to a final concentration of 2-fold and samples were incubated for 5 min at 55 °C. For comparative qualitative blotting, similar total protein amounts were separated by SDS polyacrylamide gel electrophoresis (SDS-PAGE) and transferred onto nitrocellulose membranes using a Mighty Small transfer tank (Hoefer, Holliston, MA, USA). Membranes were incubated with primary antibodies and secondary antibodies conjugated with IRDye-800 (LI-COR Biosciences, Bad Homburg, Germany), or horseradish peroxidase (HRP; Jackson ImmunoResearch, Ely, UK). In the latter case, signals were detected using SuperSignal West Pico PLUS Chemiluminescent Substrate (Thermo Fisher Scientific, Dreieich, Germany). Signals were documented using a ChemiDoc MP Imaging System (Bio-Rad). Further analyses were performed using Image Lab version 6.1.0 software (Bio-Rad).

### 2.4. Immunolocalization in Tissue Sections

For immunostaining, 6 μm thick cryosections were fixed with acetone (−20 °C) for 10 min and air-dried for 10 min. After fixation, sections were rehydrated with phosphate-buffered saline (PBS) and incubated with blocking medium (10% normal goat serum in PBS) for at least 45 min at RT. Sections were incubated with primary antibodies diluted in 1% bovine serum albumin (BSA) in PBS for 1 h at RT. After extensive washing with PBS containing 0.05% Tween 20 (PBST), sections were incubated at RT with the appropriate secondary antibodies diluted in 1% BSA in PBS for 1 h. Slides were rinsed in PBS and dipped in water before being mounted in FluoromountG mounting medium (Thermo Fisher Scientific, Dreieich, Germany), and analyzed and photographed using a Zeiss LSM710, an LSM900 confocal microscope or an AxioImager M1 microscope (all Carl Zeiss GmbH, Oberkochen, Germany). Zen 2.6 software (Carl Zeiss) was used for image processing. Controls included staining with secondary antibodies only. No unspecific staining or autofluorescence was observed.

### 2.5. Primary and Secondary Antibodies, and Other Reagents

Primary and secondary antibodies, and other reagents used for staining in this study are listed in Table 2. From the novel SYNPO2a-CT antiserum, the IgG fraction was purified from the complete serum using the Pierce ProteinA IgG purification kit (ThermoFisher Scientific) according to the instructions of the manufacturer.

### 2.6. Yeast Two-Hybrid Assays

A *SYNPO2a* cDNA fragment encoding amino acids 1 to 180, including the sequence encoding the PDZ domain, was amplified by PCR and cloned into pLexA. The plasmid was transformed into the Saccharomyces cerevisiae L40 reporter strain. Subsequently, this strain was cotransformed with a human heart cDNA library in the vector pACT2 (Matchmaker cDNA library HL4042AH; BD Biosciences Clontech, Palo Alto, CA, USA). Screening for bait interactors using selective plates without leucine, tryptophan and histidine (-LWH) and activation of the lacZ gene encoding β-galactosidase, were performed as described [25], according to the manufacturer’s protocol. Bait plasmids were isolated from the yeast cells and sequenced to identify the insert. For direct interaction assays, the *SYNPO2* bait plasmid was co-transformed with the prey plasmid to be tested.

### 2.7. Protein Expression and Purification

Recombinant proteins were expressed in *E. coli* BL21(DE3)Codon Plus cells (Stratagene, LaJolla, CA, USA). Bacteria were grown in LB medium supplemented with 100 mg/L carbenicillin and 34 mg/L chloramphenicol. Protein expression was induced at OD_600_ = 0.6 by the addition of 0.5 mM IPTG. After 3 h at 18 °C, cells were harvested by centrifugation and cell pellets were stored at −20 °C. Purification of His-tagged proteins was performed essentially as described [10,26]. Briefly, bacterial cells were lysed in lysis buffer (50 mM sodium phosphate, pH 8.0, 300 mM NaCl, 10 mM imidazole, 1 mg/mL lysozyme). DNA was fragmented by sonification and cell debris was sedimented by centrifugation (4500 rpm, 30 min, 4 °C). Soluble proteins were incubated with Ni^2+^-NTA agarose beads (Qiagen) under constant agitation at 4 °C for 1 h. Beads were washed twice with washing buffer (50 mM sodium phosphate, pH 8.0, 300 mM NaCl, 20 mM imidazole), and bound protein was eluted with elution buffer (50 mM sodium phosphate, pH 8.0, 300 mM NaCl, 250 mM imidazole). GST and GST-fusion proteins were purified using Protino Glutathione Agarose 4B according to the instructions of the manufacturer (Macherey/Nagel, Düren, Germany). Eluted proteins were stored on ice until use.

### 2.8. Co-Immunoprecipitation Assays and Western Blot Overlays

For co-immunoprecipitation experiments, 1 μg T7-tagged and 10 μg EEF-tagged purified recombinant proteins were mixed in IP buffer (0.05% Triton X-100, 1% BSA, protease inhibitors [Roche mini complete, Roche Diagnostics, Mannheim, Germany] in PBS) and incubated at room temperature (RT) for 1 h. After adding 0.7 μg of the mAb against the T7-tag, the mixture was further incubated for 30 min at RT. Subsequently, 20 μL Dynabeads protein G (Dynal Biotech, Hamburg, Germany) were added, and the mixture was incubated at 4 °C for 2 h. All previous steps were performed with gentle shaking. The beads were washed three times with IP buffer without BSA. Subsequently, they were boiled in SDS sample buffer, and bead-associated proteins were separated by SDS-PAGE, transferred to nitrocellulose membranes and, after blocking in 4% nonfat dry milk in TBST, immunodetected using antibodies directed against the respective tags and HRP-conjugated goat anti-mouse IgG + IgM, or goat anti-rat Ig. Signals were detected using SuperSignal West Pico PLUS Chemiluminescent Substrate, and documented using a ChemiDoc MP Imaging System (Bio-Rad).

For Western blot overlay experiments, purified GST or GST-tagged synemin fragments were run on a polyacrylamide gel and transferred to a nitrocellulose membrane using a Transblot SD semidry blot apparatus (Bio-Rad, Munich, Germany). Membranes were blocked for 45 min in 4% nonfat dry milk in TBST and stained with GST antibodies to verify the position of the proteins, or incubated with the bacterially expressed and purified T7-tagged PDZ domain of SYNPO2. Binding of the PDZ domain to the blotted proteins was analyzed with anti-T7-tag antibody and HRP-conjugated goat anti-mouse IgG + IgM. Signals were detected using SuperSignal West Pico PLUS Chemiluminescent Substrate and X-ray films (Fujifilm, Ratingen, Germany). Films were developed using an automatic processor for X-ray films (Curix 60, AGFA) and scanned.

### 2.9. Culture and Transfection of A7r5 Smooth Muscle Cells

A7r5 smooth muscle cells were originally obtained from the American Type Culture Collection (ATCC CRL-1444), and kindly provided by Dr. Mario Gimona (Salzburg, Austria). Cells were grown in low glucose (1 g/L) DMEM without phenol red containing 10% fetal calf serum (Sigma), 4 mM L-glutamine, 100 U/mL penicillin, 100 μg/mL streptomycin in (all from Gibco/ThermoFisher) at 37 °C and 5% CO_2_. Cells were transfected with a plasmid encoding a fusion protein of EGFP and the carboxy-terminus of human SYNPO2a (amino acids 1085-1261) using JetPRIME transfection reagent (Polyplus, Illkirch, France), following the instructions of the manufacturer. Then, 48 h after transfection, cells were fixed with 4% PFA, permeabilized with 0.5% Triton X-100, and stained with BM75.2, an antibody specific for α-actinin, and the appropriate secondary antibody (GAM IgM Cy5), while the actin cytoskeleton was stained using CoraLite594-phalloidin (Proteintech, Planegg-Martinsried, Germany). The controls included staining with secondary antibodies only. No unspecific staining was observed.

### 2.10. Densitometry

The limited availability of human tissue samples allowed for experiment optimization, with only one or two repetitions of the final experiments being feasible. Consequently, a thorough statistical analysis was deemed impracticable.

For qualitative analysis, representative agarose gels obtained from the RT-PCR experiments were subjected to densitometry using Quantity One 4.6 Software. Bands corresponding to PCR IV (all isoforms) were normalized to 1.0. The quantity of all other bands was divided by the signal of PCR IV to establish the relative quantity of each band and isoform. For PCRs I-III, representing the isoforms containing the PDZ domain (a–c), values were averaged. Relative expression levels above 80%, from 40–80%, from 10–40% and below 10% were categorized as very strong, strong, moderate and weak, respectively.

To estimate the relative abundance of various SYNPO2 isoforms in different tissues at the protein level, representative Western blots stained with HH9, our antibody recognizing all SYNPO2 isoforms, were analyzed by densitometry using Quantity One 4.6 Software. The highest protein level (i.e., that in skeletal muscle) was set to 100%, and the relative levels of SYNPO2 isoforms in other tissues were calculated. Relative expression levels above 80%, from 40–80%, from 10–40% and below 10% were classified as very strong, strong, moderate and weak, respectively.

## 3. Results

### 3.1. Tissue-Specific Expression of SYNPO2 Isoforms at the RNA Level

The gene structure and resulting mRNAs presented in Figure 1A would enable the expression of six different main SYNPO2 isoforms that we denominated SYNPO2 a–f [12]. Isoforms a, b and c are encoded by mRNAs including exons 1, 2 and a part of exon 3, all of which are lacking in isoforms d, e and f. Interestingly, exon 1 encodes a PDZ domain, the only part of SYNPO2 with a defined 3D structure. Alternative splicing at the 3’ end allows the generation of three different carboxy-termini, present in isoforms a and d, b and e, and c and f.

**Figure 1 cells-13-00085-f001:**
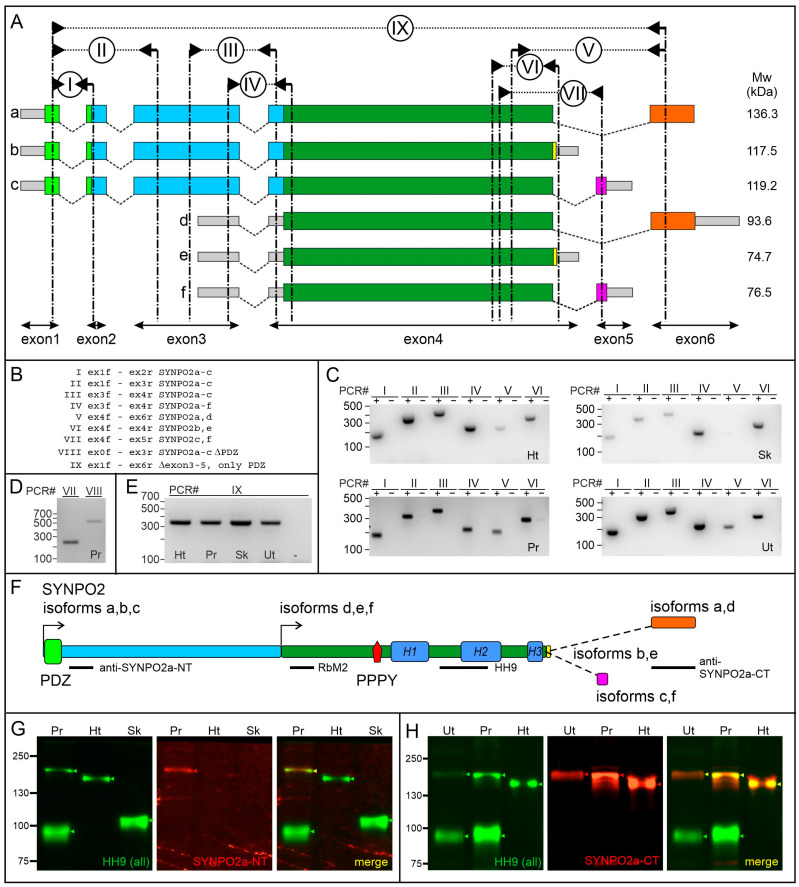
Tissue-specific expression of SYNPO2 isoforms. (**A**) Schematic presentation of the structure of the *SYNPO2* gene and splice patterns. Isoforms a/d, b/e and c/f are alternatively spliced and have different 3’ exons. Isoforms d–f start from an alternative transcription start site. Calculated molecular masses of the respective isoforms are given at the right. (**B**) Summary of the RT-PCR experiments and the *SYNPO2* isoforms amplified by the respective primer pairs. (**C**–**E**) Analysis of *SYNPO2* expression at the RNA level in different human tissues by RT-PCR. Positions of primer pairs I–IX are indicated in (**A**), except primer pair VIII, since exon 0 is not shown in (**A**). Ht: heart; Sk: skeletal muscle; Ut: uterus (smooth muscle); Pr: prostate. (**C**) The PDZ domain-containing 5’ end (I, II, III) is strongly expressed in the heart, uterus and prostate, but only very weakly in skeletal muscle. The 3’ end encoding the carboxy-terminus of isoforms a and d (V) is expressed in the uterus and prostate, but only weakly or not in both striated muscles. + and − show PCR reactions with and without addition of cDNA, respectively. (**D**) Weak expression of the exon 5-containing isoform (VII) and the isoform lacking the PDZ-domain (VIII) in prostate tissue. (**E**) An isoform in which exon 2 is directly spliced to exon 6 (IX) is detected in all tested tissues. (**F**) Graphical representation of the results of the alternative splicing events at the protein level. Two alternative transcriptional start sites (arrows) result in two distinct protein amino-termini, and inclusion of a PDZ domain only in the variants derived from the more upstream start site. Additional complexity is created at the carboxy-terminus by alternative splicing. H1, H2 and H3 indicate parts of the protein with high homology to SYNPO and SYNPO2L. The positions of epitopes of the different antibodies are indicated by the black bars below the sketch. (**G**,**H**) Analysis of SYNPO2 expression by Western blotting confirms the expression of different isoforms in different human (Pr, Ht, Sk) or mouse (Ut) muscle types and tissues. Blots were double stained with HH9 (green bands) and (**G**) SYNPO2a-NT (isoforms a/d) or (**H**) SYNPO2a-CT (isoforms a–c) as indicated (red bands). In the merged pictures, bands stained by both antibodies are marked by yellow arrowheads. Note that probably due to the high proline content of all isoforms, the proteins migrate slower than expected.

We analyzed the expression of *SYNPO2* at the RNA level in human skeletal muscle, heart, prostate and uterus by RT-PCR using primers specific for the individual exons of the human *SYNPO2* gene. Both the gene structure and the position of the respective primer pairs used are presented in Figure 1A,B. Importantly, primer pair IV, which amplifies exons, found all *SYNPO2* isoforms resulted in a strong signal in all tissues. RT-PCR using primer pairs I, II and III, which specifically amplify the exons specific for the isoforms that are extended at the 5’ end, are strongly expressed in the heart, uterus and prostate. By contrast, skeletal muscle samples yielded only very weak signals. An analysis of the splicing patterns at the 3’ end revealed that primer pair VI resulted in a signal in all tissues. By contrast, primer pair V that amplifies the splice variant containing exon 6, gave a strong signal in the uterus and prostate but only a weak signal in the heart and a hardly detectable signal in skeletal muscle (Figure 1C). Experiments using primer pairs to analyze the expression of *SYNPO2* isoforms c and f (PCR VII), and an isoform including an exon upstream of exon 1 that we named exon 0, but lacking exon1 (+Ex0Δex1) (see Appendix A), generally did not reveal conclusive results. Solely in the prostate, hardly detectable levels of these isoforms were repeatedly detected, but only when high cycle numbers were used for the PCR (Figure 1D). However, an isoform lacking exons 3, 4 and 5 (Δex3-5; PCR IX) could be readily amplified using cDNA prepared from all tissues (Figure 1E). The resulting protein (NP_001273684.1) consists of the amino-terminal 86 amino acids of SYNPO2a encoding the PDZ domain, followed by only 37 amino acids encoded by exon 6, in an alternative reading frame differing from that of *SYNPO2a*. All amplicons were of the expected length. Purification and sequencing of the amplicons confirmed their correct sequence.

A comparison of our data with similar analyses performed by Human Protein Atlas (HPA) (http://v15.proteinatlas.org/ENSG00000172403-SYNPO2/tissue (accessed on 20 January 2023)), convincingly confirms our data. Thus, skeletal muscles express mainly the *SYNPO2e* mRNA variant encompassing exons 3b and 4a, with only minimal expression of exons 1, 4b, 5 and 6. In contrast, the human heart shows high expression of exons 1, 2 and 3a, and negligible expression of exons 4b, 5 and 6, implying expression of isoform *SYNPO2b* (Appendix A). Analyses of mRNA expression in different smooth muscle-containing human tissues confirm that mRNAs encoding SYNPO2e are expressed. However, in the endometrium and prostate, these mRNAs have an extended 3’ UTR (Appendix A). This contrasts the situation in striated muscles, but also the urinary bladder and the colon, in which the 3’ UTR is considerably shorter (Appendix A), (http://v15.proteinatlas.org/ENSG00000172403-SYNPO2/tissue (accessed on 20 January 2023)). These data also show that smooth muscle cells most likely express at least two different isoforms: *SYNPO2a* (without exon 4b and including exon 6) and *SYNPO2e* (including exon 4b/c), but in different relative quantities (Appendix A). The hardly detectable expression level of exon 0 in all tissues explains the difficulty in reproducibly detecting this isoform in our RT-PCR experiments.

### 3.2. Tissue-Specific Expression of SYNPO2 Protein Isoforms

We analyzed the expression of SYNPO2 isoforms at the protein level by Western blotting using antibodies that either recognize all SYNPO2 isoforms (HH9), or are specific for the high molecular mass variants SYNPO2a–c (SYNPO2a-NT), or for SYNPO2a+e (SYNPO2a-CT), respectively. The parts of the protein used to produce the antibodies are given in Figure 1F. All SYNPO2 variants showed considerably slower gel mobility than predicted. The presence of highly acidic and basic motifs, the extremely high proline content (10.6 to 12.8%) and the fact that SYNPO2 is a highly phosphorylated protein [27], likely influence the hydrodynamic size of the proteins or limit binding of SDS. Both human striated muscle types seem to predominantly express one major detectable isoform (Figure 1G,H). Together with our data obtained by RT-PCR (see above), and the observation that the heart isoform is stained by the antibody SYNPO2a-NT that is specific for SYNPO2a–c, but not with anti SYNPO2a-CT, we conclude that the heart contains mostly SYNPO2b, but skeletal muscle displays mainly isoform SYNPO2e (Figure 1G,H). In mouse uterus and human prostate tissue, the expression pattern is somewhat more complicated: both contain a high proportion of smooth muscle cells, and at least two different isoforms are expressed. In uterus and prostate, the smaller isoform seems to have a smaller molecular mass than the isoform in skeletal muscle, and most likely represents SYNPO2e (Figure 1G,H). The lower molecular mass protein expressed in skeletal muscle that runs slower does not represent isoform SYNPO2d, since it is not stained by anti-SYNPO2a-CT (Figure 1G), and might represent posttranslationally modified (possibly phosphorylated) SYNPO2e [7,27,28]. The higher molecular mass protein in the uterus and prostate runs slower than the cardiac isoform, and includes the amino acids encoded by exon 6, as revealed by staining with anti SYNPO2a-CT. Unfortunately, antibodies specific for SYNPO2c+f, the PDZ domain (NP_001273684.1), and the amino acids encoded by exon 0, are not available.

This differential expression of SYNPO2 isoforms with and without PDZ domain in different muscle cells is confirmed by stainings performed by HPA (v23). Staining with antibodies recognizing either all SYNPO2 isoforms or only SYNPO2a–c indicates the highest expression of the PDZ domain-containing isoforms in the heart (Appendix A). In contrast, skeletal muscle and smooth muscle cells in the endometrium, prostate, kidney and colon show strong staining with an antibody that recognizes all SYNPO2 isoforms, while an antibody specific for the PDZ-containing isoforms SYNPO2a–c shows weak or no staining (Appendix A). Especially, the staining of mesangial smooth muscle-like cells in the glomeruli of the kidney reveals the sole expression of a SYNPO2 isoform without the PDZ domain (Appendix A). Densitometric analyses of the above described RT-PCR and Western blotting results are summarized in Appendix A and Table 3.

An analysis of the expression levels of SYNPO2 in a panel of mouse striated muscles indicated considerable differences in the expression level of a single, short isoform (probably SYNPO2e) in individual skeletal muscles (Appendix A). In the mouse heart, only low levels of both, a short isoform (probably SYNPO2e) and a long isoform (SYNPO2b) were detected. In the mouse uterus, large quantities of a short isoform with identical mobility to that in striated muscles, and low levels of SYNPO2a, an isoform larger than the one expressed in the heart, were found (Appendix A).

### 3.3. The PDZ-Domain of SYNPO2 Interacts with Synemin

Previously, we identified VPS18 as an interaction partner of the amino-terminal PDZ domain of SYNPO2 isoforms a, b and c (Figure 2A) through a yeast two-hybrid screening of a human heart cDNA library [13]. In the same screen, four prey clones contained a partial cDNA coding for synemin isoforms A and B (Figure 2B). All clones included the carboxy-terminus of synemin. Since PDZ domains typically interact with the carboxy-termini of their ligands, we investigated whether mutagenesis or deletion of this part of synemin inhibited the interaction between both proteins. Indeed, direct yeast two-hybrid assays confirmed that substitution of the ultimate or penultimate amino acids (---GHWF to ---GHAF, or ---GHWR, respectively) resulted in a complete loss of the ability to interact: yeast cells with the mutant variants did not activate the reporter genes necessary for growth on selective plates (Figure 2C), and induction of ß-galactosidase activity (Figure 2D). These data were confirmed biochemically using Western blot overlay experiments. While the PDZ domain of SYNPO2 bound a fusion protein of the carboxy-terminus of synemin (amino acids 1464-1565) with GST (GST-Synm-CT), it did not bind a variant lacking the carboxy-terminal 32 amino acids (GST-Synm-CTΔ) or GST (Figure 2E).

To further support the interaction of SYNPO2 and synemin, we investigated whether both proteins colocalize in human smooth muscle cells and cardiomyocytes. Immunolocalization in cryosections of the human colon and heart revealed that SYNPO2 and synemin both localize in dense bodies of the smooth muscle cells of the muscularis externa of the colon (Figure 2F), and in the Z-disc region of cardiomyocyte myofibrils, identified by staining for filamin C (Figure 2G). Notably, SYNPO2 is absent from intercalated discs where its binding partner filamin C is localized.

### 3.4. The Isoform-Specific Part of SYNPO2 Encoded by Exon 6 Binds α-Actinin

SYNPO2 isoforms a and d contain a carboxy-terminal extension that is encoded by exon 6 (Figure 1A and Figure 3A). Since our RT-PCR data indicated that this part of SYNPO2 is mainly expressed in smooth muscle, but not in striated muscle, we concentrated on its localization in smooth muscle cells. Transient transfection of solely the part of SYNPO2 that is encoded by this exon fused to GFP (SYNPO2a-CT-GFP) indicated that this SYNPO2 fragment is sufficient for its association with the phalloidin-stained actin cytoskeleton (Figure 3B, upper panel). Higher magnification revealed that SYNPO2a-CT-GFP colocalized with the well-characterized dense body component α-actinin in stress fiber dense bodies (Figure 3B, lower panel). To investigate whether SYNPO2a CT can directly interact with α-actinins, co-immunoprecipitation experiments with bacterially expressed and purified EEF-tagged α-actinin-1 or α-actinin-2 and T7-tagged SYNPO2a-CT were performed. In both cases, precipitation of T7-SYNPO2-CT with anti-T7 tag antibodies led to co-precipitation of α-actinin, indicating a direct interaction of both proteins.

### 3.5. SYNPO2 Is Expressed in Uterus Smooth Muscle Cells

Our RT-PCR and Western blotting data (see Figure 1) indicate expression of two different isoforms in the human uterus: one low molecular mass isoform, probably identical to the isoform expressed in skeletal muscle, and a second variant that probably includes both, the amino-terminal and the carboxy-terminal extensions. To analyze in which kind of cells SYNPO2 is expressed in the uterus, cryosections of mouse uterus were co-stained with antibodies HH9 (recognizing all SYNPO2 isoforms) and BM75.2 (detecting all α-actinin isoforms). Already at low magnification, it was apparent that SYNPO2 is strongly expressed in the myometrium, but not in the endometrium that contains uterine glands and small blood vessels embedded in connective tissue (Figure 4A). Higher magnification unequivocally revealed that the smooth muscle cells of the myometrium (and not surrounding connective tissue) express SYNPO2 (Figure 4B,C).

### 3.6. SYNPO2 Is Mainly Localized in Smooth Muscle Cells and Not in Epithelial Cells of the Human Prostate

Because of published data indicating that deletions in the *SYNPO2* gene were associated with invasive prostate cancer [29], we aimed at revealing the specific localization of SYNPO2 in the normal human prostate. Staining of cryosections with antibodies against all SYNPO2 isoforms (RbM2) and against all cytokeratins revealed that epithelial cells identified by keratin staining were not stained by anti-SYNPO2 antibodies (Figure 5A,B). By contrast, anti-smooth muscle actin and anti-SYNPO2 stained exactly the same cells, indicating that SYNPO2 is highly expressed solely in smooth muscle cells, whereas glandular epithelial cells contain no detectable SYNPO2 protein (Figure 5C).

### 3.7. SYNPO2 Is a Component of Nemaline Rods and the Central and Intermediate Zone of Target Fibers

Based on the presence of multiple α-actinin binding sites in SYNPO2 [10], and the fact that α-actinin is a well-characterized nemaline rod protein, we hypothesized that SYNPO2 might be a component of nemaline rods. To test this, cryosections of skeletal muscles from nemaline myopathy patients were double stained with antibodies against both proteins. Indeed, SYNPO2 and α-actinin colocalized in areas showing an abnormal sarcoplasmic distribution of α-actinin (Figure 6A).

In addition, our previous work indicated that SYNPO2 is overrepresented in the central and intermediate zone of target fibers in the skeletal muscles of patients with neurogenic muscular atrophy [20]. Co-staining of cryosections with antibodies against SYNPO2 and filamin C (a marker for these zones) confirmed the overrepresentation of SYNPO2 in these structures when compared to normal fibers without targets (Figure 6B).

## 4. Discussion

The early expression of SYNPO2 during muscle differentiation had implied an important function of this protein, at least for cross-striated muscles [4,10]. This view was reinforced by the strong, deleterious effects of the knockdown of SYNPO2 expression upon applying mechanical stress, again in striated muscles [12], but also in smooth muscle cells [13]. The identification of *Synpo2* to be a lethal gene in mice [15] indicates the vital importance of SYNPO2 at the whole animal level, at least for mouse embryo development. The position of the inserted loxP sites (https://www.alliancegenome.org/allele/MGI:4441661 (accessed on 20 September 2023)) points to a knockout strategy aimed at a deletion of the constitutive exon3. A precise analysis of the effects on the expression of the individual *Synpo2* isoforms was not performed, but based on our gene and transcript analyses, this deletion should affect the expression of all isoforms a-f, indicating that the reported embryonic lethality cannot be attributed to the lack of only one or more individual isoforms. The cardiovascular system was one of the systems described to be most severely affected in heterozygous mice, together with the hematopoietic and immune systems (https://www.informatics.jax.org/allele/MGI:5548760 (accessed on 20 September 2023)). Thus, it is highly probable that the prime reason for lethality in homozygous mice is malfunction of the developing heart. This is in agreement with the expression of *Synpo2* in the mouse heart from Theiler stage 15 (E9-10.25) onwards (https://www.informatics.jax.org/gxd/marker/MGI:2153070 (accessed on 20 September 2023)). Unfortunately, no detailed description of the mouse phenotype is available. Additional evidence supporting the importance of SYNPO2 comes from ClinVar (https://www.ncbi.nlm.nih.gov/clinvar (accessed on 22 September 2023)), which lists 52 different *SYNPO2* variations in humans, which are all described as either “uncertain significance” or “benign”, but no data concerning potential associated phenotypes are currently available. The few variations that have been reported so far to be true human disease candidates, indeed result in a phenotype (in this case monogenic nephrotic syndrome, proteinuria) when they are homozygous [16,17]. Our work therefore paves the way for future detailed analyses of mouse models and/or patients.

### 4.1. SYNPO2 Isoform Expression in Striated Muscles

Previous studies identified the existence of multiple SYNPO2 isoforms with distinct functions [8,9,10,11,12], while their potentially different expression in cells and tissues has not been investigated. We show here for the first time detailed tissue-specific, differential expression of multiple SYNPO2 isoforms. Table 3 summarizes our findings and conclusions about specific isoform expression. The comparison of our data with those from HPA (http://v15.proteinatlas.org/ (accessed on 20 January 2023)) confirm that skeletal muscles express mainly an mRNA variant with exons 3b, 4 and 4a, but no expression of exons 1, 2, 3a, 4b, 5 and 6 (*SYNPO2e*). In contrast, the human heart shows an additional high expression of exons 1, 2 and 3a, compatible with the sole expression of a PDZ domain-containing isoform (SYNPO2b). Additionally, in the heart, exon6 is hardly expressed.

An analysis of the expression levels of SYNPO2 in various mouse muscles revealed considerable differences in the abundance of a single, short isoform in individual skeletal muscles (Appendix A). Proteomic analysis of single mouse muscle fibers has demonstrated consistent SYNPO2 expression levels across both fast and slow fiber types [30], indicating that variations observed in skeletal muscles cannot be solely attributed to fiber type predominance. Similarly, human skeletal muscles have shown no significant differences in SYNPO2 expression levels across different muscle fiber types [31]. In the mouse heart, there are comparable levels of a short isoform, likely SYNPO2e, and the longer isoform SYNPO2b. However, these levels are much lower when compared to most skeletal muscles. In the mouse uterus, high levels of a short isoform with identical mobility to that in striated muscles were observed, along with low levels of an isoform that is larger than that in the heart, most likely SYNPO2a (Appendix A). Future investigations on specific tissues and cell types should consider these findings. In this context, it will also be crucial to assign specific functionalities to individual SYNPO2 isoforms. 

### 4.2. The PDZ Domain of SYNPO2 Isoforms a–c Also Interacts with Synemin: An Additional Novel Link between Z-Discs and the Surrounding Intermediate Filaments

We previously described that the PDZ domain of SYNPO2 interacts with vacuolar protein sorting 18 homolog (VPS18) in smooth muscle cells, establishing a link between SYNPO2 and autophagosome formation [13]. Interestingly, SYNPO2 variants containing the PDZ domain are minimally detectable in skeletal muscle, but are abundant especially in the heart and in specific smooth muscle cells. The distinct functions of SYNPO2 isoforms a,b,c therefore appear to be most important in these cell types. Similarly, the novel interaction between SYNPO2 and synemin isoforms A and B that we describe in this work may not be essential in skeletal muscle fibers, although also in skeletal muscle, both proteins are localized in the Z-disc region (see Figure 2 and refs. [4,10,32]). In this tissue, the interaction between synemin and α-actinin [33,34,35], together with plectin that provides an additional link of intermediate filaments (IFs) with the Z-disc, may provide sufficient stability for muscle function. In cardiomyocytes, a continuously active striated muscle cell type, the interaction between SYNPO2 and synemin represents an additional link between the desmin intermediate filament cytoskeleton surrounding the myofibrils and the Z-disc. This interaction could relieve the permanently mechanically challenged Z-discs and enhance the stability of the contractile apparatus of the cardiomyocytes, necessary to resist continuous contractions. The significance of the interaction of SYNPO2 with the intermediate filament system is further highlighted by the recently identified direct interaction between a specific Z-disc-associated plectin isoform and SYNPO2 [36].

### 4.3. SYNPO2 in Smooth Muscles

Immunoelectron microscopy revealed that SYNPO2/fesselin is a constituent of dense bodies within the smooth muscle cells of the chicken gizzard [5]. Dense bodies contain filamin and are associated with both microfilaments and IFs [37], and their function resembles that of Z-discs in striated muscle cells. In smooth muscle cells, synemin colocalizes with desmin IFs [32,38]. Via its direct interaction with microfilaments and actin-binding proteins such as α-actinins and filamins, and its indirect interaction with IFs via synemin and plectin, SYNPO2 assumes a pivotal role in stabilizing the contractile apparatus of smooth muscle cells. Intriguingly, a diminished expression of both SYNPO2 and synemin has been observed in quiescent smooth muscle cells that have lost their contractile properties after vascular injury, as well as in atherosclerotic plaques [39]. Furthermore, the expression of both the *SYNPO2* and *SYNM* genes was sharply upregulated in tracheoesophageal fistulas when compared to normal esophagus and trachea [40]. This indicates that both genes are co-regulated during alterations of the phenotype of smooth muscle cells.

While striated muscle cells mainly display a single SYNPO2 isoform, both prostate and uterus, i.e., organs rich in smooth muscle cells, express at least two different isoforms. Interestingly, we could only convincingly detect the expression of an isoform featuring an additional α-actinin-binding site encoded by exon 6 (SYNPO2a, but not SYNPO2d) in smooth muscle cells. However, the relative amount of this isoform was, when compared to the expression level of SYNPO2e, higher in the prostate than in the uterus. This implies the expression of SYNPO2a in smooth muscle cells with a specific function and/or phenotype. Transcript expression data from HPA support this observation: the endometrium reveals similar levels of exon 4b- and exon 6-containing isoforms, compatible with expression of isoforms a and d, whereas smooth muscle cells of the prostate hardly express exon 6, mainly indicating the expression of SYNPO2e. Such isoform-specific differences in expression levels have rarely been described previously [41]. The bladder, for example, contains far more SYNPO2 than the aorta. Also, the relative amount of the low molecular mass isoform (presumably SYNPO2e) compared to the expression level of SYNPO2a was reported to be higher in the bladder [41]. This not only indicates that the expression of SYNPO2 may vary in general, but also that the expression of individual isoforms is subject to complex regulation in smooth muscle cells of different tissues, as well as smooth muscle cells of different developmental and pathophysiological stages [41]. Similar to the proposed important function during striated muscle development and maintenance, SYNPO2 was suggested to be a marker of contractile smooth muscle cells, and to be involved in smooth muscle cell differentiation and function. Its reduced expression upon vascular injury, regulated by the level of actin polymerization, and its deregulation in vascular disease are further indicators for such a role [41]. In summary, all these and our novel findings reveal that SYNPO2 expression is highly regulated in smooth muscle cells, with SYNPO2a predominantly expressed in a specific subset of smooth muscle cells. We propose that the presence of the synemin-binding PDZ domain and the additional α-actinin binding site in SYNPO2a may contribute to a protein interaction network that is associated with functional differences of specific smooth muscle cells that require increased stabilization. This observation aligns with the known involvement of alternative splicing in several other RNAs playing a crucial role in smooth muscle cell plasticity [42].

### 4.4. The 5’ UTR of Individual SYNPO2 mRNAs Is Extremely Long in Smooth Muscle Cells

Analyses of mRNA expression in different human tissues conducted by HPA (https://v15.proteinatlas.org (accessed on 20 January 2023)) indicate that mRNAs encoding SYNPO2b and/or SYNPO2e, are expressed in various tissues with distinct 3’ UTRs, due to alternative polyadenylation. In striated muscles, for example, the 3’ UTR is considerably shorter than in smooth muscle-containing tissues such as urinary bladder, colon, prostate. A comparison of cDNAs NM_001389264.1 and NM_001128933.3 reveals that the latter is more than 7000 bp longer, whereas the predicted encoded proteins have identical carboxy-termini. This might have a major impact on post-transcriptional control by miRNAs and RNA-binding proteins. Furthermore, it is well known that 3’ UTRs may regulate mRNA stability, direct mRNA localization to subcellular regions and impact translational control [43,44]. cDNA NM_001128933.3 would even include the predicted exon 5. In this respect, it is interesting to note that in several human cell lines, the simultaneous expression of *SYNPO2* isoforms a, b and c was detected, whereas specific antibodies failed to detect any SYNPO2 protein [8], implying a strong impact of translational regulation on SYNPO2 expression. A comparative analysis of mRNA and protein expression levels, as presented in Table 3, supports this notion.

### 4.5. Further Predicted Isoforms

The isoform complexity of *SYNPO2* might be further increased by the existence of an isoform starting from an alternative transcription start site upstream of exon 1 in Figure 1A. Apparently, this exon 0 is spliced to exon 2 (+ex0Δex1 in Appendix A), implying that isoforms a, b and c might also exist without a functional PDZ domain that is in part encoded by exon 1. The loss of the PDZ domain would hinder this isoform from interacting with synemin in the intermediate filament system, but also with VPS18, which would also lead to a loss of its connection with autophagosome formation during CASA [13]. Since our data, as well as the data from HPA, indicate none to very low levels of the mRNA encoding this protein in all tested tissues, it is at this point still relatively unlikely that this variant plays a major role in striated and smooth muscle cells. A separate database entry (cDNA AK304121; protein BAG65020) describes an mRNA variant, in which exon 2 is directly spliced to exon 6 (Δex3,4,5 in Appendix A), resulting in the deletion of exons 3 and 4. This variant was a result of a human cDNA sequencing project and directly submitted to the database. The encoded 13.4 kDa protein consists of the complete PDZ domain plus 37 amino acids encoded by exon 6 in a different reading frame than in SYNPO2a/d. Our RT-PCR experiments confirmed the expression of this mRNA variant in all tissues. The expression of essentially only the PDZ domain without the rest of the SYNPO2 protein that contains several binding sites for other proteins might block binding of the full length, PDZ domain-containing SYNPO2 isoforms a, b and c to both, synemin and VPS18, potentially attenuating both pathways. Future investigations are required to investigate whether both protein variants, and especially the 13.4 kDa variant, truly exist in certain cell types, and whether they impact vital cell functions by affecting CASA or the stability of dense bodies and Z-discs.

Expression of *SYNPO2c/f* was detected at the mRNA level in the human prostate. Interestingly, and different from all other exons (except exon 0), base-wise conservation analysis in 100 vertebrates by PhyloP (https://genome-euro.ucsc.edu (accessed on 20 August 2023)) indicates that exon 5 is the evolutionarily least conserved of all exons (Appendix A). A Blast search (https://blast.ncbi.nlm.nih.gov/Blast.cgi (accessed on 20 August 2023)) at the protein level indicates high homology solely within primates, and only modest homology with seals, narwhal and bears, but no homology with other vertebrates.

### 4.6. SYNPO2 in Skeletal Muscle Pathology

In a previous study using immunolocalization of several myofibrillar proteins in myofibrillar myopathy (MFM) patients, we found that in MFM patients, some Z-disc proteins such as α-actinin, titin, SYNPO2/myopodin and SYNPO2L/tritopodin were rather stable Z-disc components, whereas other proteins (filamin C, myotilin, ZASP, Xin and XIRP2) revealed striking alterations in these pathological situations [45]. These data were later confirmed by our proteomic studies on protein aggregates in the muscle fiber of MFM patients: whereas the “stable” proteins in the first group were not overrepresented in aggregates, the second set of more mobile proteins was highly increased in these structures [46,47]. Since the SYNPO2 binding partner α-actinin is a component of nemaline bodies, we reasoned that SYNPO2 might also be found in these pathological structures. Indeed, in two different patients with nemaline myopathy, SYNPO2 colocalized with α-actinin in the rods. These findings are in agreement with a recent study that analyzed nemaline rod proteins by laser capture microdissection and mass spectrometry, and found SYNPO2 among the proteins that were overrepresented in nemaline rods [48].

Similarly, we also found overrepresentation of SYNPO2 in the target fibers of skeletal muscles of patients with neurogenic muscular atrophy [20]. This finding was confirmed by our immunolocalization studies, demonstrating that both SYNPO2 and filamin C are components of these targets (see Figure 6). Since SYNPO2 is also a component of Z-bodies of nascent myofibrils appearing early in muscle cell differentiation, this observation supports our conclusion that in these targets, regeneration processes take place [20]. This suggests that the underlying pathophysiology is more similar in nemaline myopathy and neurogenic muscular atrophy, but distinct from protein aggregation diseases like MFM.

### 4.7. SYNPO2 in Cancer

Although we have focused our analyses on SYNPO2 expression in muscle cells, it is important to also discuss our data in the context of the prostate and its pathologies. This is particularly justified by a number of publications hinting at the reduced expression of SYNPO2 in a variety of tumors, leading to the assumption that it may act as a tumor suppressor (e.g., ref. [29,49,50,51]). Analysis of SYNPO2 expression in normal prostate and prostate cancer specimens by HPA using two different antibodies showed the expression of SYNPO2 protein only in smooth muscle cells, both in normal tissue and in more than 20 tumor specimens (https://www.proteinatlas.org/ENSG00000172403 SYNPO2/pathology/prostate+cancer#imid_19955532 (accessed on 22 September 2023)) [52,53]. These data confirm and complement the results of our immunolocalization studies in the human prostate, but are in sharp contrast to an in situ hybridization study using a probe specific for all *SYNPO2* isoforms, indicating its exclusive expression in prostate epithelium [29]. However, single cell analysis performed by HPA [54] shows very high *SYNPO2* expression in smooth muscle cells and high expression in fibroblasts, but only low expression in glandular epithelial cells (https://www.proteinatlas.org/ENSG00000172403-SYNPO2/single+cell+type/prostate (accessed on 22 September 2023)). These results are confirmed both by immunolocalization data presented in this work, and the immunolocalization data of HPA (see also Appendix A). Collectively, it is very likely that simply the lack of smooth muscle cells in high-grade prostate tumors [49] may cause the apparent absence of SYNPO2 protein in prostate (and other) tumor biopsies. In addition, gross changes in SYNPO2 isoform expression by different splicing pathways and/or translational control may impact the proposed tumor suppressor activity of SYNPO2 during tumor formation. 

## 5. Conclusions

In this study, we have verified the expression of SYNPO2 in skeletal, cardiac and smooth muscle cells. Furthermore, we have demonstrated that SYNPO2 isoforms exhibit a tissue-specific expression pattern. More specifically, the amino-terminal PDZ domain of SYNPO2 isoforms a, b and c is predominantly expressed in cardiac and smooth muscle cells, where it interacts with the carboxy-terminus of the intermediate filament protein synemin. This interaction establishes an additional connection between intermediate filaments and Z-discs in cardiac muscle cells and dense bodies in smooth muscle cells, respectively. Notably, SYNPO2 isoform a, which contains a carboxy-terminal extension, is primarily expressed in smooth muscle cells and not in striated muscle. This extension binds α-actinins and is sufficient for targeting dense bodies when expressed in smooth muscle cells. While skeletal and cardiac muscle cells express only one SYNPO2 isoform, smooth muscle cells express two different variants, which might depend on their differentiation state and contractility. Our data contribute to elucidating specific functions of distinct isoforms not only in different adult muscle types but also during various developmental stages of muscle cells. The intricate network of interactions underscores the diverse roles played by SYNPO2 in different muscle types, highlighting its adaptability to the mechanical demands of specific cellular environments. Our analyses lay the foundation for future functional studies that will closely investigate the precise roles of individual isoforms.

## Figures and Tables

**Figure 2 cells-13-00085-f002:**
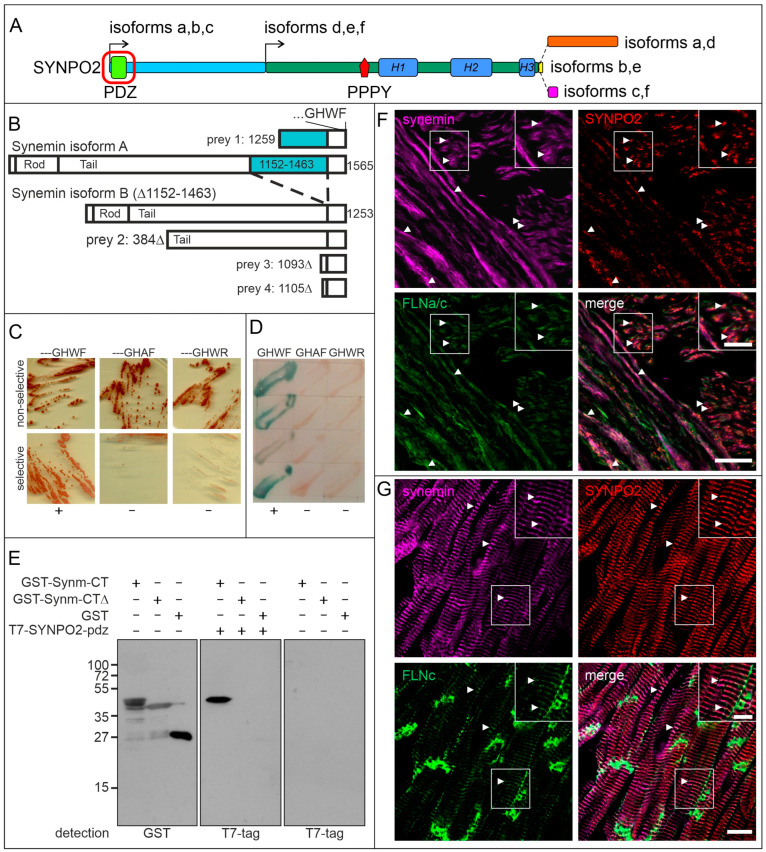
The PDZ domain of SYNPO2 interacts with synemin. (**A**) Schematic illustration of SYNPO2 isoforms illustrating the position of the PDZ domain in isoforms a, b and c (red box). (**B**) A yeast-two-hybrid screen identifies synemin isoforms A (prey 1) and B (preys 2, 3, 4) as novel binding partners of this domain. “GHWF” are the amino acids at the C-terminus of the preys. (**C**,**D**) Mutagenesis of the carboxy-terminus of synemin, reveals specific interaction of SYNPO2 with the carboxy-terminus of wildtype synemin (---GHWF), but not with the mutant variants (---GHAF, and ---GHWR). (**E**) Western blot overlay assays indicate interaction of SYNPO2 with GST-synemin amino acids 1464-1565, but not with carboxy-terminally truncated synemin (GST-Synm-CTΔ, amino acids 1464-1533) and GST. Left panel: localization of the recombinant proteins revealed by staining with anti-GST antibody; center panel: overlay with SYNPO2-PDZ showing interaction with GST-Synm-CT only; right panel: incubation with T7-antibody without overlay with SYNPO2-PDZ. (**F**,**G**) Immunofluorescence localization of SYNPO2 and binding partners in sections of smooth and striated muscle samples. (**F**) SYNPO2 and synemin partially colocalize in dense bodies in the muscularis externa of human colon (arrowheads). (**G**) SYNPO2 and synemin colocalize with filamin C (FLNc) in the Z-disc region of human cardiomyocytes (arrowheads). Insets in (**F**,**G**) show a magnification of the respective boxed areas. Bars: 10 μm. 5 μm (insets).

**Figure 3 cells-13-00085-f003:**
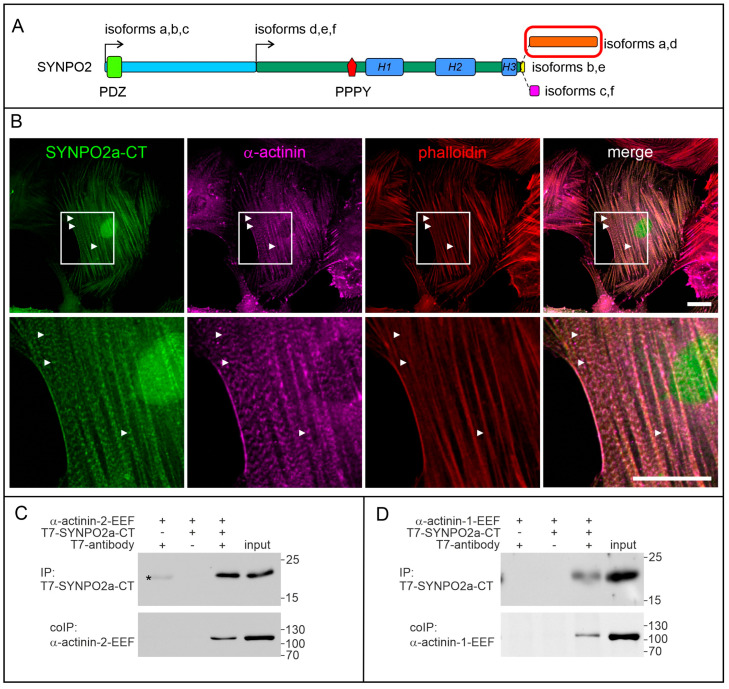
The carboxy-terminal extension of SYNPO2a and d localizes to dense bodies in smooth muscle cells and interacts with α-actinin. (**A**) Schematic illustration of SYNPO2 isoforms illustrating the position of the carboxy-terminal extension of isoforms a and d (red box). (**B**) Transient expression of the carboxy-terminal extension of SYNPO2 isoforms a and d (SYNPO2a-CT) fused to GFP in A7r5 smooth muscle cells shows targeting of the fusion protein to dense bodies that were identified by staining for α-actinin (arrowheads). The boxed area in the upper panel is shown enlarged in the lower panel. Bars: 20 μm. (**C**,**D**) Co-immunoprecipitation experiments reveal binding of the carboxy-terminal extension of isoforms a and d to α-actinin-2 (**C**) and α-actinin-1 (**D**). The weak signal marked with an asterisk indicates the light chain of the T7-tag antibody used for the immunoprecipitation.

**Figure 4 cells-13-00085-f004:**
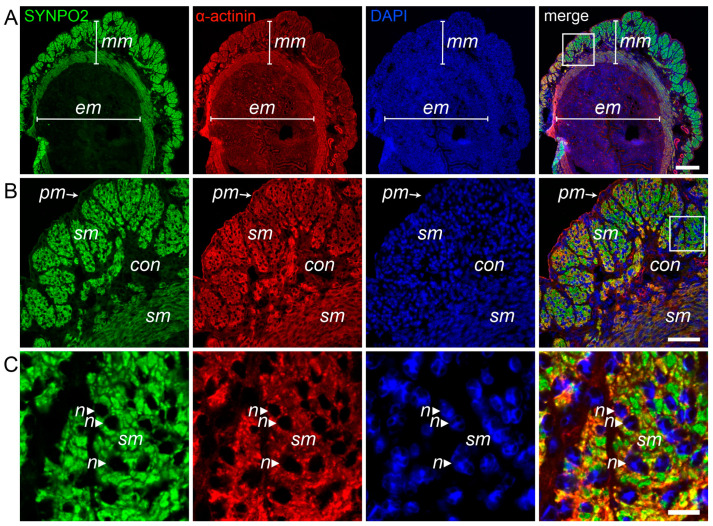
SYNPO2 is expressed in smooth muscle cells of the uterus. (**A**–**C**) Immunolocalization of SYNPO2 and α-actinin in cryosections of mouse uterus. DAPI was used to stain nuclei. (**A**) SYNPO2 is strongly expressed in the myometrium (mm), but not in the mucosa of the endometrium (em) that contains uterine glands and arteries embedded in connective tissue stroma. (**B**) Higher magnification of the boxed area in (**A**) reveals strong expression of SYNPO2 in smooth muscle (sm) cells of the myometrium, but not in surrounding connective tissue (con) and the outer connective tissue layer, the perimetrium (pm). (**C**) Enlargement of the boxed area in B shows specific and exclusive SYNPO2 staining of smooth muscle cells. Also note that nuclei (n, arrows) are not stained. Bars: 200 μm (**A**), 50 μm (**B**), 10 μm (**C**).

**Figure 5 cells-13-00085-f005:**
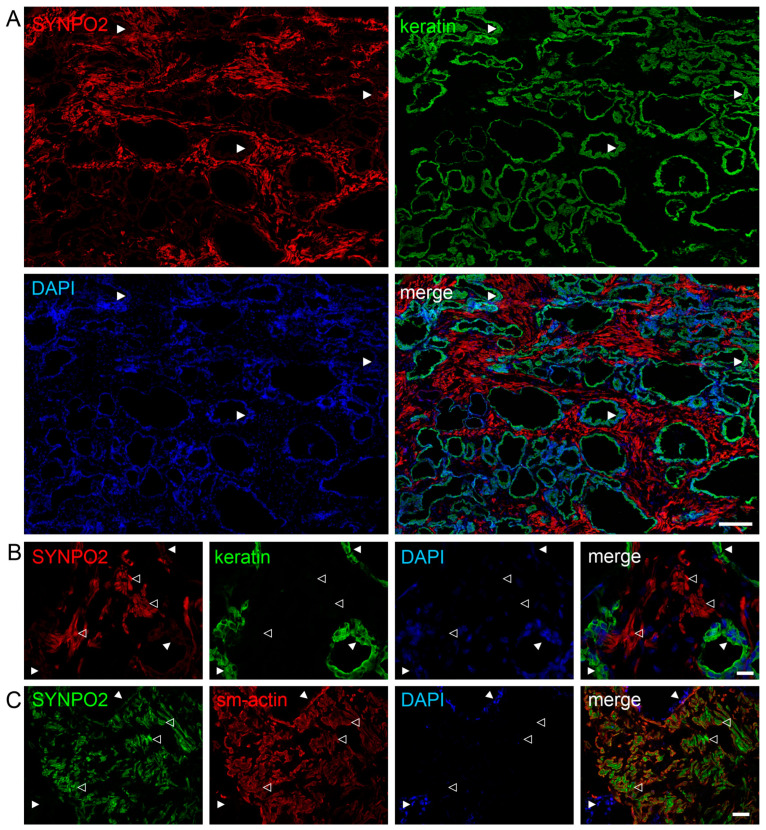
SYNPO2 is mainly localized in smooth muscle cells and not in epithelial cells of the human prostate. (**A**–**C**) Immunolocalization of SYNPO2 and keratin (**A**,**B**) or smooth muscle actin (sm-actin, **C**) in cryosections of human prostate. DAPI was used to stain nuclei. (**A**) Low magnification reveals strong expression of SYNPO2 in stromal cells, but no detectable staining in epithelial cells, which in turn are identified by staining for keratin (arrowheads). (**B**) Higher magnification confirms the absence of SYNPO2 in keratin-positive glandular epithelial cells (closed arrowheads). Instead, stromal cells are stained (open arrowheads). (**C**) Staining for sm-actin unequivocally clarifies that it is smooth muscle cells that are strongly stained by the SYNPO2 antibody (open arrowheads), whereas epithelial cells are not stained (closed arrowheads). Bars: 200 μm (**A**), 20 μm (**B**,**C**).

**Figure 6 cells-13-00085-f006:**
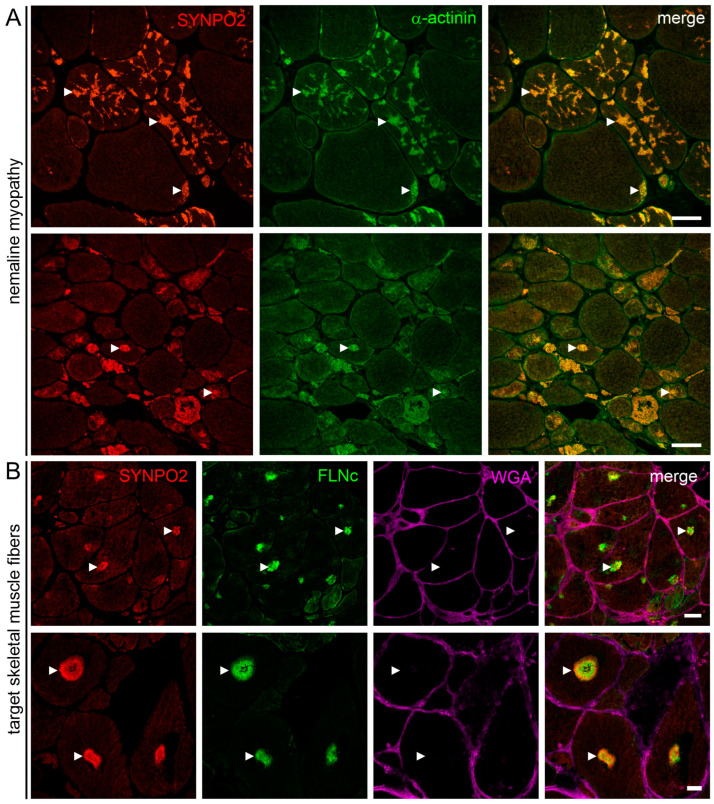
SYNPO2 is a component of nemaline bodies and the central and intermediate zone of target fibers. (**A**) Skeletal muscle cryosections of two patients with nemaline myopathy stained with antibodies against SYNPO2 and α-actinin. SYNPO2 is found together with the verified rod component α-actinin in nemaline bodies (arrowheads). (**B**) Skeletal muscle cryosections of two patients with neurogenic muscular atrophy stained with antibodies against SYNPO2 and filamin C (FLNc). SYNPO2 localizes together with filamin C in the center of the targets that occur in the muscle fibers of these patients (arrowheads). Wheat germ agglutinin (WGA) was used to visualize the borders of the muscle fibers. Bars: 50 μm (**A**), 20 μm (**B**).

**Table 1 cells-13-00085-t001:** Oligonucleotides used in this study. I-IX refer to the RT-PCR experiments indicated in Figure 1A,B.

RT-PCR	Forward Primer	Reverse Primer
I	ATGACTGGAGGGGCGCCCTG exon1	AGCTTGATGACTTCAGGGTA exon2
II	ATGACTGGAGGGGCGCCCTG exon1	CTGCAGGGTGGTACTTTCCA exon3
III	TGTCCACATAAATTCGATCC exon3, 5′	ACTTTCACTCCTCCTGAGCC exon4
IV	AGCAGACCTCACAAGCACCG exon3, 3′	CCTCTCGCTCAAGCTCGCCA exon 4
V	TCAGTCAAGGTCAATTCAGCC exon4	TCATCCACTAGACCGAGAGG exon 6
VI	TTCTCAGCCAAGAAAAGTGG exon4	TCCACAACAGATGGTTTCCA exon 4
VII (exon 5)	GCAGGCTTCGTCAGTGTACT exon4	AGGCTGGGAAGTAGGTCCTT exon 5
VIII (+ex0/Δex1)	GAATTCGGCAACTCGGAAGC exon0	CTTTCAAGCTGCCTGCACAG exon 3
IX (Δex2,3,4)	ATGACTGGAGGGGCGCCCTG exon1	TCATCCACTAGACCGAGAGG exon 6

**Table 2 cells-13-00085-t002:** Primary and secondary antibodies, as well as further reagents, were used in this study to label specific proteins or compartments. The antigens used to raise SYNPO2 antibodies are specified (amino acid numbers refer to the human SYNPO2a isoform NP_597734.2). GAM: goat anti-mouse; GAR: goat anti rabbit; GARat: goat anti rat; AF: Alexa Fluor; Ig: immunoglobulin; WGA: wheat germ agglutinin; DAPI: 4′, 6-Diamidino-2-phenylindole dihydrochloride. Dilutions of the applied antibodies and reagents are given. IF: immunofluorescence; WB: Western blotting.

Antibody/Reagent	Target	Species	Source and/or Reference	Dilution
HH9	aa 803-916; SYNPO2a-f	mouse IgG1	[10]	IF 1:2; WB 1:50
RbM2	aa 428-534; SYNPO2a-f	rabbit	[10]	IF 1:1000
anti-SYNPO2a-CT	aa 1085-1262; SYNPO2a,d	rabbit	custom made, BioGenes, Berlin, Germany	WB 1:1000
anti-SYNPO2a-NT	aa 66-151; SYNPO2a,b,c	rabbit	Atlas antibodies, Bromma Sweden, HPA049707	WB 1:500
BM75.2	α-actinin	mouse IgM	Sigma-Aldrich Chemie, Taufkirchen, Germany A5044	IF 1:75
RR90	filamin A/C	mouse IgA	[23]	IF 1:50
anti-SYNM	synemin	rabbit	Atlas antibodies HPA040066	IF 1:200
C11	cytokeratins (pan)	mouse IgG1	Sigma-Aldrich Chemie, C2931	IF 1:400
1A4	smooth muscle actin	mouse IgG2a	Sigma A2547	IF 1:600
YL1/2	EEF-tag	rat IgG2a	ThermoFisher Scientific, Dreieich, Germany, MA1-80017 [24]	WB 1:700
T7-tag	T7-tag	mouse IgG2b	Sigma-Aldrich Chemie 69522	WB 1:10,000
GST-tag	GST	mouse IgG1	Sigma-Aldrich Chemie, 71097	WB 1:10,000
GAM IgA AF488	mouse IgA	goat	SouthernBiotech, Birmingham, AL, USA, 1040-30	IF 1:40
GAM IgG2a AF594	mouse IgG2a	goat	ThermoFisher Scientific A-21135	IF 1:500
GAM IgG1 AF594	mouse IgG1	goat	Jackson ImmunoResearch, Ely, UK, 115-585-205	IF 1:300
GAM IgM AF546	mouse IgM	goat	ThermoFisher Scientific A-21045	IF 1:100
GAM IgM Cy5	mouse IgM	goat	Jackson ImmunoResearch 115-175-075	IF 1:100
GAR Cy3	rabbit Ig	goat	Jackson ImmunoResearch 111-165-045	IF 1:800
GAR AF647	rabbit Ig	goat	ThermoFisher Scientific A-21245	IF 1:100
GAR-HRP	rabbit IgG	goat	Jackson ImmunoResearch 111-035-144	WB 1:10,000
GAM-HRP	mouse IgG + IgM	goat	Jackson ImmunoResearch 115-035-068	WB 1:10,000
GARat-HRP	rat IgG + IgM	goat	Jackson ImmunoResearch 112-035-068	WB 1:10,000
GAR IRdye800CW	rabbit	goat	LI-COR Biosciences, Bad Homburg, Germany 926-32211	WB 1:10,000
CoraLite594- phalloidin	actin filaments	-	Proteintech, Planegg-Martinsried, Germany, PF00003	1:100
WGA Oregon Green 488	glycoproteins, cell membrane	-	ThermoFisher Scientific W6748	1:250
DAPI	nuclei	-	Sigma-Aldrich Chemie, D9542	1:40,000

**Table 3 cells-13-00085-t003:** Summary of SYNPO2 isoform expression in human tissues. +++: very strong expression; ++: strong expression; +: moderate expression; ±: weak expression; -: no expression. * no specific antibody available for this isoform. ^#^ For Western blotting, protein extract from mouse tissue was used. Evaluation was performed as described in Section 2.10.

	RNA (RT-PCR)					Protein (Western Blotting)			
SYNPO2 Isoforms	Skeletal Muscle	Cardiac Muscle	Prostate	Uterus	SYNPO2 Isoform	Skeletal Muscle	Cardiac Muscle	Prostate	Uterus ^#^
a–c	+	+++	+++	++	a	-	-	+	±
a–f	+++	+++	+++	+++	b	-	++	-	-
a,d	±	+	++	+	c *				
b,e	+++	++	+++	++	d	-	-	-	-
c,f	-	-	±	-	e	+++	-	++	+
					f *				

## Data Availability

Anonymized data not published within this article will be shared upon request from qualified investigators.

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
