# Peer review of "Synaptopodin-2 Isoforms Have Specific Binding Partners and Display Distinct, Muscle Cell Type-Specific Expression Patterns"

_cells, 2023, doi:10.3390/cells13010085_

Round 1

Reviewer 1 Report

Comments and Suggestions for Authors

This manuscript investigates the existence of several alternative splicing variants from the SYNPO2 gene and the expression of Synaptopodin-2 isoforms in - mainly- different types of muscle tissue. In addition, the authors demonstrate an interaction between Synaptopodin-2 variants that contain a PDZ domain at the N-terminus with synemin, an intermediate filament protein and show that Synaptopodin-2 is also a component of nemaline bodies in skeletal muscle myopathies.

This is a solid piece of work and the data, while mainly descriptive, are of high quality. It paves the way for future functional studies that will investigate the exact role of the individual isoforms more closely by knockdown experiments. I have just a few comments that should help to clarify some aspects.

  1. Throughout the manuscript they refer to “skeletal muscle”. Obviously there are fast and slow twitch types of skeletal muscle, which may be different in Synaptopodin-2 isoform expression. This is partially supported by the data in Fig S3 from mouse, where most of mainly fast twitch (GN, QUAD, TA) have higher levels of Synpo2e than the slow twitch (DIA, SOL) with the exception of EDL. Access to different types of human skeletal muscle is obviously limited, so I do not expect them to test that, but it should be stated in the legend, which muscle type was studied (looks like GN for the PC & Westerns?) to give an indication of expected fiber type dominance and the issue should also be discussed in the text. 
  2. In the description of Figure 2 it is important to stress that Synaptopodin-2 seems to be completely absent from the intercalated disc.
  3. Panels C and D appear to be swapped in Figure 2.
  4. Fig S3 suggests extremely low expression of Synaptopodin-2 in mouse heart compared to other muscle tissue, while in the human heart gets three crosses for most isoforms. Can the authors speculate why that is the case?
  5. Label in Fig S3 should be Synpo2 to indicate murine.
  6. In the discussion correlate PDZ-containing Synaptopodin-2 expression levels with Synemin expression levels for different tissues?
  7. Line 600: Also the data from Table 3 suggests this - refer to it!
  8. Line 679: “Authors should discuss the results….” - this seems to be superfluous and needs deleting.

Author Response

This manuscript investigates the existence of several alternative splicing variants from the SYNPO2 gene and the expression of Synaptopodin-2 isoforms in - mainly- different types of muscle tissue. In addition, the authors demonstrate an interaction between Synaptopodin-2 variants that contain a PDZ domain at the N-terminus with synemin, an intermediate filament protein and show that Synaptopodin-2 is also a component of nemaline bodies in skeletal muscle myopathies.

This is a solid piece of work and the data, while mainly descriptive, are of high quality. It paves the way for future functional studies that will investigate the exact role of the individual isoforms more closely by knockdown experiments. I have just a few comments that should help to clarify some aspects.

We thank Reviewer 1 for the very positive comments.

  1. Throughout the manuscript they refer to “skeletal muscle”. Obviously there are fast and slow twitch types of skeletal muscle, which may be different in Synaptopodin-2 isoform expression. This is partially supported by the data in Fig S3 from mouse, where most of mainly fast twitch (GN, QUAD, TA) have higher levels of Synpo2e than the slow twitch (DIA, SOL) with the exception of EDL. Access to different types of human skeletal muscle is obviously limited, so I do not expect them to test that, but it should be stated in the legend, which muscle type was studied (looks like GN for the PC & Westerns?) to give an indication of expected fiber type dominance and the issue should also be discussed in the text.

We have now included information about the muscles the human biopsies were taken from (lines 105&106). Furthermore, we now cite two papers (Murgia et al., Skelet Muscle. 2021;11(1):24 and Eggers et al., Proteomes. 2021;9(2):28.), describing that there is no significant difference of SYNPO2 expression in different fiber types in mice and humans, respectively. (lines 537-541).

  1. In the description of Figure 2 it is important to stress that Synaptopodin-2 seems to be completely absent from the intercalated disc.

We thank the reviewer for this comment and added the sentence "Notably, SYNPO2 is absent from intercalated discs, where its binding partner filamin C is localized.." to the manuscript (lines 389&390).

  1. Panels C and D appear to be swapped in Figure 2.

Thank you for pointing out this mistake. We have corrected Figure 2 accordingly.

  1. Fig S3 suggests extremely low expression of Synaptopodin-2 in mouse heart compared to other muscle tissue, while in the human heart gets three crosses for most isoforms. Can the authors speculate why that is the case?

We have adapted the number of crosses for the human heart that gets two crosses now. This is in agreement with the expression level mentioned by the Human Protein Atlas: moderate for heart and high for skeletal and smooth muscle.

  1. Label in Fig S3 should be Synpo2 to indicate murine.

This would be correct in case of gene names (e.g. Igf1 for mouse and IGF1 for human). However, according to the Protein Nomenclature Guidelines the correct proteins designations for mice and human are the "same as the gene symbol, but not italicized and all upper case (e.g.: IGF1)".

  1. In the discussion correlate PDZ-containing Synaptopodin-2 expression levels with Synemin expression levels for different tissues?

Figure S1 indicates rather low expression levels of SYNPO2 variants containing the exons encoding the PDZ domain in skeletal and smooth muscles. In the heart the relative amount of PDZ-containing SYNPO2 isoforms is somewhat higher. This does not correlate to the apparent expression levels of synemin: synemin expression is highest in skeletal muscle, although no PDZ-containing SYNPO is expressed. Therefore, we think that without true quantitative data it is very hard to discuss this topic in depth. This is beyond the scope of this work and we therefore prefer to refrain from it.

  1. Line 600: Also the data from Table 3 suggests this - refer to it!

We thank the reviewer for this suggestion. We now refer to this table in the Discussion (Line 630).

  1. Line 679: “Authors should discuss the results….” - this seems to be superfluous and needs deleting.

These lines have been deleted. It obviously is a remnant of the journal template that we forgot to delete.

Reviewer 2 Report

Comments and Suggestions for Authors

In the current study, the authors evaluated the expression pattern and the binding partners of SYNPO2 isoforms.

This is an interesting study, nevertheless, it required significant improvements.

Questions/ suggestions/ limitations of the study.

Abstract:

Line 26: “Furthermore, it was proposed to play a role in tumor cell proliferation and metastasis.” Is this relevant for the striated muscle which has very low cancer/metastasis rates? Also, you have no experimental data in this manuscript on tumor cell proliferation and metastasis.

Line 24: “Synaptopodin-2 (SYNPO2) is a Z-disc-associated component of striated muscle cells.”Our analyses at mRNA and protein levels revealed differential expression of SYNPO2 isoforms in cardiac, skeletal, and smooth muscle cells”. You cannot state that SYNPO2 is a component of striated muscle cells if it is documented to be expressed in smooth muscle. Please revise.

Introduction:

Line 51: “In the latter work, it was confirmed that the protein is expressed in all types of muscle cells and localizes to Z-discs in striated muscle cells”. Add smooth muscle localization.

Line 60: Figure 1. Is this your original Figure or is it modified from the previous publications? If so, you need to acknowledge the previous publications in the Figure legend.

Materials and Methods:

Line 96: Please provide the Human study approval number and date, as well as the description of how the informed consent was obtained.

Line 96: Please describe how the oligonucleotides were designed.

Lines 135, 145, and 159: Please provide the dilutions of all primary and secondary antibodies used.

Line 218: Please provide the source of the A7r5 cell line.

Please add the statistical evaluation section.

Results:

Line 286: Please clearly describe that Figure S1 represents data from the Human Protein Atlas.

Figures 2F and 2G: Please provide higher magnification images as inserts showing examples of co-localization areas indicated with the arrowheads.

Figure 3B: Alpha-actinin staining looks continuous and does not reflect the dot-like distribution of dense bodies below the sarcolemma of smooth muscle cells. How was the specificity of the staining with this antibody verified?

Figure 4: Alpha-actinin staining in this figure more resembles actin staining as can be evaluated by the distribution and intensity. Please clarify.

Figure 6: Please provide a control staining with a secondary antibody only to verify that this is a specific staining and not autofluorescence of the damaged skeletal muscle fibers.

The conclusion is very weak and does not adequately summarize the results of this study. It needs to be rewritten.

Comments on the Quality of English Language

Minor editing of English language required.

Author Response

In the current study, the authors evaluated the expression pattern and the binding partners of SYNPO2 isoforms.

This is an interesting study, nevertheless, it required significant improvements.

We thank the reviewer for finding our study interesting. We edited English language, and believe that the manuscript has improved.

Questions/ suggestions/ limitations of the study.

Abstract:

Line 26: “Furthermore, it was proposed to play a role in tumor cell proliferation and metastasis.” Is this relevant for the striated muscle which has very low cancer/metastasis rates? Also, you have no experimental data in this manuscript on tumor cell proliferation and metastasis.

In this case "tumor cell proliferation and metastasis" refers to many different tumors and not to muscle tumors. We added the sentence "in many different kinds of cancer" (line 28) to make this clear.

Line 24: “Synaptopodin-2 (SYNPO2) is a Z-disc-associated component of striated muscle cells.” “Our analyses at mRNA and protein levels revealed differential expression of SYNPO2 isoforms in cardiac, skeletal, and smooth muscle cells”. You cannot state that SYNPO2 is a component of striated muscle cells if it is documented to be expressed in smooth muscle. Please revise.

With stating that SYNPO2 is a component of striated muscle cells, we do not claim that is it exclusively expressed in these cells. In fact, SYNPO2 is expressed in many different, also non-muscle cell types. To circumvent this potential mistake, we now mention in the Abstract that "In smooth muscle cells SYNPO2 is a component of dense bodies." (line 26&27)

Introduction:

Line 51: “In the latter work, it was confirmed that the protein is expressed in all types of muscle cells and localizes to Z-discs in striated muscle cells”. Add smooth muscle localization.

In the work cited here expression, but no exact localization of SYNPO2 in smooth muscle cells was reported. We have added reference to another paper describing the localization of SYNPO2 in smooth muscle (line 55).

Line 60: Figure 1. Is this your original Figure or is it modified from the previous publications? If so, you need to acknowledge the previous publications in the Figure legend.

Although this Figure has unavoidable similarities to many other published figures describing SYNPO2 isoforms, it is new and original.

Materials and Methods:

Line 96: Please provide the Human study approval number and date, as well as the description of how the informed consent was obtained.

This information was already presented in a special section at the end of the original manuscript. Nevertheless, we added in the M&M section (lines 115-118): "Use of the material was approved by the ethical committees of the Universities of Cologne/Bonn (#13-091), the Ruhr-University Bochum (#4368-12) and the University of Giessen (#AZ07/09). Written informed consent was obtained from all subjects involved in this study."

Line 96: Please describe how the oligonucleotides were designed.

We added to paragraph 2.2 (lines 126-130) "Oligonucleotides were designed using OligoPerfect Primer Designer (https://www.thermofisher.com/de/de/home/life-science/oligonucleotides-primers-probes-genes/custom-dna-oligos/oligo-design-tools/oligoperfect.html) by submitting the sequence of the regions flanking the exon boundaries."

Lines 135, 145, and 159: Please provide the dilutions of all primary and secondary antibodies used.

In the revised manuscript we have now included the applied antibody dilutions in Table 2.

Line 218: Please provide the source of the A7r5 cell line.

In the revised manuscript we now state that A7r5 cells were "originally obtained from the American Type Culture Collection (ATCC CRL-1444), and a kind gift of Dr. Mario Gimona (Salzburg, Austria)." (lines 231-233)

Please add the statistical evaluation section.

Our work was mainly qualitative and with low sample numbers. Therefore, use of statistical evaluations was not required.

Results:

Line 286: Please clearly describe that Figure S1 represents data from the Human Protein Atlas.

To make this point clear, we have added to the legends of this figure "as presented by the Human Protein Atlas v15."

Figures 2F and 2G: Please provide higher magnification images as inserts showing examples of co-localization areas indicated with the arrowheads.

In the revised manuscript we now provide the requested enlarged insets.

Figure 3B: Alpha-actinin staining looks continuous and does not reflect the dot-like distribution of dense bodies below the sarcolemma of smooth muscle cells. How was the specificity of the staining with this antibody verified?

Proliferating A7r5 cells do not have the typical spindle-shaped morphology of smooth muscle cells in tissue. They do, however, contain many actin bundles that contain so called stress fiber dense bodies appearing like pearls on a string. These dense bodies were stained with BM75.2 a widely used, well characterized pan a-actinin antibody. Specificity of the staining was verified by omission of the primary antibodies. We state this now in the manuscript (lines 242-243).

Figure 4: Alpha-actinin staining in this figure more resembles actin staining as can be evaluated by the distribution and intensity. Please clarify.

For these stainings we clearly used  a-actinin antibodies, but neither actin-specific antibodies or phalloidin. We applied a well-characterized and widely used a-actinin antibody and we have no doubt about its specificity. It has to be kept in mind that in panels A and B we present overview pictures of whole uterus in a relatively thick section that were photographed in a "conventional" fluorescence microscope. With these limitations in mind and at the magnification shown, the figures present standard results for a-actinin stains in tissues. The panels in Figure 4C are higher magnifications, revealing the expected dot-like a-actinin staining.

Figure 6: Please provide a control staining with a secondary antibody only to verify that this is a specific staining and not autofluorescence of the damaged skeletal muscle fibers.

We thank the reviewer for this suggestion. Of course, the necessary controls with omission of primary antibodies are part of our daily routine and were made. Since we didn't see any non-specific staining, we did not see the need to take any pictures. To make this point clear, we now state in the Materials and Methods section: "Controls included staining with secondary antibodies only. No unspecific staining or autofluorescence was observed." (lines 164-165)

The conclusion is very weak and does not adequately summarize the results of this study. It needs to be rewritten.

In the revised manuscript we have rephrased the Conclusions section.

Reviewer 3 Report

Comments and Suggestions for Authors

In the current study the authors investigated the distribution of Synaptopodin-2 isoforms in different tissues, mainly human muscle tissue. The data show a prominent role for the protein in the organization of the contractile apparatus.

Major comment:

Although all experiments are well performed and of interest the presentation of the data requires attention.

Fig. 1: Please arrange the figure in the order as the different parts are presented. In other words part 1e must be Fig. 1b etc.

Fig. 2 and Table 3: There is no information how the authors decided about the quantification in weak, strong, very strong etc. Please provide representative examples as supplement. This is required to give the reader a chance to judge about these data.

Fig. 4: The description in lines 412-423 is properly for Fig. 4 not 5.

The abstract gives the reader the impression that the link between intermediate and Z-disc is a novel link. If this is indeed the first description of such a relationship it should be part of the title.

Author Response

In the current study the authors investigated the distribution of Synaptopodin-2 isoforms in different tissues, mainly human muscle tissue. The data show a prominent role for the protein in the organization of the contractile apparatus.

Major comment:

Although all experiments are well performed and of interest the presentation of the data requires attention.

Fig. 1: Please arrange the figure in the order as the different parts are presented. In other words part 1e must be Fig. 1b etc.

We thank the reviewer for pointing this out. We have rearranged the panels accordingly, to present the parts in the correct order.

Fig. 2 and Table 3: There is no information how the authors decided about the quantification in weak, strong, very strong etc. Please provide representative examples as supplement. This is required to give the reader a chance to judge about these data.

The data summarized in Table 3 is based on the data depicted in Figure 1. It has to be kept in mind that these are no quantitative RT-PCRs and also the western blots are only semi-quantitative. We therefore decided to use only these coarse categories. After reanalyzing the data, we reduced the SYNPO2 expression in the  human heart from "strong" (+++)  to moderate (++), which is in agreement with analyses performed by the Human Protein Atlas (v15). Here the SYNPO2 protein expression level is described as "high" in skeletal and smooth muscle cells, and "moderate" in cardiomyocytes.

Fig. 4: The description in lines 412-423 is properly for Fig. 4 not 5.

We thank the reviewer for pointing out this mistake, which we have corrected.

The abstract gives the reader the impression that the link between intermediate and Z-disc is a novel link. If this is indeed the first description of such a relationship it should be part of the title.

Association of intermediate filaments with Z-discs has been described before. The link between synemin and SYNPO2 is novel, though. Therefore, we changed the sentence in the abstract to "SYNPO2 therefore represents an additional and novel link between intermediate filaments and the Z-discs..." (lines 36&37).

Round 2

Reviewer 2 Report

Comments and Suggestions for Authors

The authors addressed most of my critiques in the revised manuscript. Nevertheless, I am convinced that the Statistical Evaluation section has to be added to this manuscript. Yes, most of the data is descriptive and is not quantified. The data in Table 3 provide data description as “+++: strong expression; ++: moderate expression; +: weak expression; ±: very weak expression; -: no expression”. The minimum that has to be done is to provide a clear description of how this evaluation was done, how many samples were analyzed for each assay, and how many times the assays were repeated.

The conclusion is still very vague. “we have analyzed the expression patterns of the different SYNPO2 isoforms ... and revealed tissue-specific expression of individual isoforms.” Add a few clear statements listing 2-3 most critical/novel things that your study showed so the readers will understand the essential contribution of your paper to the existing body of knowledge.
